# Degradable Slow-Release Fertilizer Composite Prepared by Ex Situ Mixing of Inverse Vulcanized Copolymer with Urea

**Ali Shaan Manzoor Ghumman** [1,2], **Rashid Shamsuddin** [1,2,*], **Mohamed Mahmoud Nasef** [3], **Carmelo Maucieri** [4,*], **Obaid Ur Rehman** [1], **Arief Aizat Rosman** [1], **Mohamed Izzat Haziq** [1] **and Amin Abbasi** [1]

1   Chemical Engineering Department, Universiti Teknologi PETRONAS, Seri Iskandar 32610, Perak Darul Ridzuan, Malaysia; ali_19001079@utp.edu.my (A.S.M.G.); obaid_19000977@utp.edu.my (O.U.R.); arief.aizat_25017@utp.edu.my (A.A.R.); izzatq98@gmail.com (M.I.H.); amin_18000407@utp.edu.my (A.A.)
2   HICoE, Centre for Biofuel and Biochemical Research (CBBR), Institute of Self-Sustainable Building, Universiti Teknologi PETRONAS, Seri Iskandar 32610, Perak, Malaysia
3   Department of Chemical and Environmental Engineering, Malaysia Japan International Institute of Technology, Universiti Teknologi Malaysia, Jalan Sultan Yahya Petra, Kuala Lumpur 54100, Malaysia; mahmoudeithar@cheme.utm.my
4   Department of Agronomy, Food, Natural Resources, Animals and Environment-DAFNAE, Agripolis Campus, University of Padua, Viale dell'Univerità 16, 35020 Legnaro, PD, Italy
*   Correspondence: mrashids@utp.edu.my (R.S.); carmelo.maucieri@unipd.it (C.M.)

**Abstract:** To improve the crop yield and nitrogen uptake efficacy, a novel slow-release urea composite fertilizer (SUCF) was developed using inverse vulcanized copolymer with better biodegradation and nutrient release longevity. Copolymers were synthesized via inverse vulcanization of jatropha oil, and their properties were evaluated using thermogravimetric analysis (TGA), Fourier transform infrared spectroscopy (FTIR), powdered-X-ray diffractometry (p-XRD), and scanning electron microscopy (SEM). SUCFs were developed by ex situ mixing of inverse vulcanized copolymer with urea powder using mechanical mixer, and their properties were evaluated using Fourier transform infrared spectroscopy (FTIR) and scanning electron microscopy (SEM). FTIR spectra of developed fertilizer possesses the urea characteristics peaks along with the undisturbed peaks representing copolymer, confirming the mechanical mixing and that no reaction took place. SEM images of the SUCFs compared with images of copolymer revealed the appearance of new isolated particles with different morphology; EDX mapping showed that these particles represent the urea added to the copolymer. Nitrogen release longevity of developed fertilizers was evaluated in both soil and distilled water. The leaching test revealed that only 70% of the total nitrogen of SUCF prepared from 50 wt% sulfur copolymer was released after 16 days of incubation in distilled water, whereas it released only 35% nitrogen after 20 days in soil. The biodegradability of all copolymers developed was investigated by burying in soil and it revealed their biodegradable nature as weight loss was observed, which increased with the increase of incubation period.

**Keywords:** slow-release fertilizers; fertilizer matrix; urea; inverse vulcanization; sulfur polymers

## 1. Introduction

It is projected that global population, which is 7.9 billion today, will exponentially grow to 10 billion by 2050 [1,2]. Therefore, to make food security certain for this constantly increasing population, by increasing the crop yield while reducing the environmental impact and preserving the soil health, will be challenging. The agriculture sector is consuming ever larger amounts of the fertilizers to boost the crop yield, especially nitrogen (N) fertilizers, which have so far shown adverse consequences [3,4]. Urea is the most used N fertilizer. However, in the soil, the N is vulnerable to losses due to surface run-off, nitrate leaching, and ammonia volatilization, thus disturbing the neighboring ecosystem [5,6]. Nearly 70% of all urea applied to crops is estimated to be lost to the environment, resulting in low nutrient-use efficiency (NUE) and high cost [7–9].

Agricultural experts and enterprises have been attempting to create solutions to this growing challenge to obtain agronomic and environmental advantages by developing innovative fertilizers, also defined as "smart fertilizers", which allow control over the rate, timing, and duration of nutrient release [10]. Slow-release fertilizers (SRFs) are intentionally produced fertilizers that delay nutrient release to synchronize this with crop nutrient requirements, enhancing crop production and NUE [11]. SRFs have been developed using a variety of materials, including synthetic and natural polymers as well as inorganic materials. Although synthetic polymers have shown encouraging results in terms of nutrient release longevity, the use of hazardous solvents in the coating of synthetic polymers on urea, as well as their nonbiodegradability, pollutes the environment and soil [12–15]. Natural polymers, on the other hand, suffer from their hydrophilic character, which causes nutrients to be released abruptly and for an unexpected period [16]. The brittle nature of inorganic materials, such as sulfur (S), encourages the formation of micropores on the coated surface, resulting in nutrient release failure [4,17–19].

Recently, hydrogels-based SRFs have attracted the attention of researchers, and so numerous biopolymers have been utilized to produce them. SRF hydrogels are produced by entrapping the nutrients in the matrix of as-synthesized hydrogels. Despite the numerous advantages of such SRFs, they are still suffering from several challenges, such as the timing of fertilizer decomposition, inhibition of polymerization process, precise control of polymerization rection conditions, excess usage of methanol requiring the removal of byproducts, and low nutrient release longevity. Such intimidating factors lead to the need to look for other green, sustainable, and biodegradable materials and methods to produce SRFs.

Inverse vulcanized copolymers are a new class of stable S-enriched polymers produced using a simple and solvent-free method [20–24]. Inverse vulcanization is a recently developed technique that makes it feasible to produce S-based polymers that are firm against depolymerization of polymeric chains over time [25,26]. Pyun et al. showed that adding a comonomer, such as 1,3-diisopropenylbenzene (DIB), with one or more vinylic bonds to the ring-opened elemental S can result in highly stable S-based polymers [26]. In other words, inverse vulcanization suggests that adding a petroleum-based or bio-based comonomer to the S melt can stabilize the S chains into the polymeric structure, preventing it from depolymerization. So far, most of the monomers used in inverse vulcanization are petroleum-based; nevertheless, some bio-based monomers, and also several vegetable oils, have also been studied in the production of S-based polymers [24,27]. These vegetable oils include oils of rubber seed [22,23], corn [21], olive [28], sunflower [28], linseed [28], algae [29], canola [30], and soybean [31]. S-based polymers have shown a wide range of properties and potential applications depending on the monomer type, composition, and the reaction conditions [32]. Inverse vulcanization suggests a whole novel application for elemental S, directly reducing the environmental disadvantages of the vast quantities of elemental S openly left and piled up as a byproduct in gas and petroleum refineries [33,34]. Besides this, the usage of vegetable oils as sustainable comonomers in inverse vulcanization can contribute to the green chemistry.

Almost 80 wt% of jatropha oil consists of unsaturated fatty acids which, together with the fact that this oil is a nonedible oil, make jatropha oil a remarkable choice for inverse vulcanization [35,36]. Jatropha oil mainly consists of unsaturated fatty acids, i.e., oleic acid (34.3 to 45.8 wt%) and linoleic acid (29.0 to 44.2 wt%), depending on the growing conditions and the jatropha species [35,37]. In addition, S is a secondary, yet indispensable, nutrient required for plant growth. Valle et al. [31] have reported that inverse vulcanized copolymer has potential to improve the S oxidation, hence providing $SO_4^{-2}$ in a more convenient way than to elemental S. The high S content, better S oxidation, and the biodegradable nature of these copolymers have attracted and planted a seed for this research. With this in mind, the aim of this study was to synthetize a novel SRF by ex situ mixing of the inverse vulcanized copolymer produced from jatropha oil using a mechanical mixer.

## 2. Materials and Methods

The inverse vulcanized copolymer was synthesized using jatropha oil and was then characterized using Fourier transform infrared spectroscopy (FTIR), thermogravimetric analysis (TGA), powdered X-ray diffraction (p-XRD), and scanning electron microscopy (SEM). Produced SRF was also characterized using FTIR and SEM, and the N release of the SRF was studied in distilled water and soil. The kinetics data was evaluated using four different kinetic models. Soil burial test was conducted to investigate the biodegradability of copolymers.

### 2.1. Materials

Potassium chloride (AR grade), phenylmercuric, diacetyl monoxime (DAM-AR grade), thiosemicarbazide (TSC), phosphoric acid, and sulfuric acid were purchased from Sigma-Aldrich, St. Louis, MO, USA. Sulfur (AR grade) was procured from PC laboratory reagents, Malaysia. Jatropha oil was acquired from Kinetics Chemicals SDN BHD, Malaysia. All materials were used as received and no further purification was carried out.

### 2.2. Methods

#### 2.2.1. Synthesis of Copolymers

S was heated to a temperature of 170 °C, using thermoset oil bath, in a 30 mL glass vial under continuous stirring to start the ring-opening process. The process of ring opening is accompanied by the color change; as S melts, it turns into a yellow colored liquid followed by turning into an orange colored liquid which confirms the completion of the ring-opening process. As the process was completing, jatropha oil was added to a glass vial in a dropwise manner to avoid sudden drop of temperature. After this, the mixture was allowed to react for 1 h under vigorous stirring [22,23].

#### 2.2.2. Characterization of Copolymers

Fourier Transform Infrared Spectroscopy (FTIR)

FTIR analysis of copolymers was carried out to confirm the formation of polymer using a PerkinElmer frontier model spectrometer coupled with attenuated total reflectance (ATR). The scanning range for the analysis was 500–4000 cm$^{-1}$ frequency with 4 cm$^{-1}$ resolution. In total, eight scans were carried out to obtain the spectra of all copolymers.

Thermogravimetric Analysis (TGA)

TGA analysis was performed to investigate the thermal stability of the copolymer using a PerkinElmer STA 6000 simultaneous thermal analyzer. The temperature range was 25–800 °C with 10 °C min$^{-1}$ heating range under N$_2$ atmosphere.

Powdered X-ray Diffraction (p-XRD)

Structural properties of copolymers were evaluated using p-XRD analysis using a Malvern PANalytical X'Pert powder diffractometer coupled with copper anode material at 40 mA with K$_\alpha$1 = 1.540598 Å and K$_\alpha$2 = 1.544426 Å in a transition geometry using a capillary spinner.

Scanning Electron Microscopy (SEM)

A Zeiss EVO LS 15 microscope armed with Oxford Instruments INCAx-act EDX spectroscope was used to study the morphological properties of copolymers after coating with gold using a sputter coater (Emitech K550X).

#### 2.2.3. Synthesis of Slow-Release Fertilizer Composites

SRF composited were synthesized by ex situ (in different pots) mixing of produced copolymers with powdered urea using mechanical mixing at a speed of 150 rpm to obtain matrix with uniform distribution of the urea, and consequently N. The mixing was performed immediately after removing the glass vial from heating after 1 h for 10 min. The

SRF was then allowed to cool at room temperature. Equal mass of the copolymer and urea powdered was mixed. All three produced copolymers were mixed to produce three different fertilizer composites and were named as SUCF-50 (fertilizer with 50 wt% S), SUCF-60 (with 60 wt% S), and SUCF-70 (with 70 wt% S). All obtained SUCFs were analyzed using FTIR to confirm the successful mixing of the urea and to confirm no reaction took place.

### 2.2.4. Characterization of Produced SRFs

SRFs were characterized using SEM and FTIR technique to investigate the morphology and chemical structural properties, respectively. The procedure to carry out these analyses was similar to that reported in Section 2.2.2.

### 2.2.5. Nitrogen Release Test in Distilled Water

Prior to the leaching test, the evaluation of the total N content in all the SRF composites was conducted employing the Kjeldahl method [38]. Then, $2.0 \pm 0.1$ g of each of the composites was inserted in a 250 mL Erlenmeyer flask containing 200 mL of distilled water. The flask was then capped using cling film to avoid the vaporization of water.

To estimate the amount of leached N in the water, 2.5 mL of the stirred aliquot was taken out and the water was replaced with 200 mL of fresh distilled water. A mixture of the obtained aliquot and 7.5 mL of the color reagent in a 60 mL glass vial was then placed in a thermostatic water bath at a temperature of 85 °C for 30 min in order to generate the red-colored solution. The concentration of the urea in each sample controls the red color intensity of that sample. After 30 min, the solutions were left to be cooled in 20 °C tap water for 20 min. The overall release time was calculated using the triplicates and the standard curve method.

The DAM solution and the TSC solution were produced by dissolving 2.5 g of DAM in 100 mL of distilled water and 0.25 g of TSC in 100 mL of distilled water, respectively, while 10 mL of sulfuric acid, 250 mL of phosphoric acid, and 240 mL of distilled water were combined to form the acid reagent. Consequently, 15 mL of TSC solution, 25 mL of DAM solution, and 460 mL of the acid reagent were mixed to form the colored reagent solution [39].

### 2.2.6. Nitrogen Release in Soil

Soil (sand 20.5%, silt 39.3%, and clay 40.2%) was obtained from a local nursery and was sieved using a 2 mm sieve to remove debris and roots, followed by air drying to 20–30% moisture content. SRFs were placed under soil and the moisture content maintained throughout the experiment. After every 24 h, the soil was taken and mixed with 100 mL of 2 M KCl–PMA solution to extract the urea. The mixture was allowed to mix at 160 rpm speed for 1 h, wrapped with tinfoil. After extraction, the solution was filtered to remove the soil using Whatman microfiber glass filter paper. After filtration, 2.5 mL of aliquot was taken from the filter solution and mixed with 7.5 mL of color reagent following a similar procedure to that explained in Section 2.2.5.

### 2.2.7. Soil Burial Test

In order to examine the biodegradability of the copolymer, a soil burial test was carried out. To conduct this, 2 g of the copolymer was enfolded with a teabag-shaped mesh bag and was buried into the soil (depth of 10 cm) in a polymer container. During the experiment, the soil was kept moisturized. The mesh bag was taken out after each regular interval and was washed using distilled water to ensure no soil remained on it. It was then dried using an oven with a temperature of 60 °C to obtain a constant weight. Accordingly, the equation below was used to determine the weight loss of the copolymer [40].

$$Weight\ loss\ (\%) = \frac{m_1 - m_2}{m_1} \times 100 \tag{1}$$

where $m_1$ was the initial weight and the $m_2$ the final weight.

## 3. Results and Discussion

The copolymers, synthesized by reacting jatropha oil and S using inverse vulcanization process, were brown in color and rubbery in nature. The intensity of darkness of the color decreased with increase of initial S content and led to brittle material. FTIR spectra of all obtained copolymers along with jatropha oil is depicted in Figure 1. By analyzing and comparing the spectra, it was found that the spectrum of the pristine jatropha possesses two cis-alkene characteristic peaks which appear at 1660 and 3009 cm$^{-1}$, representing stretching vibrations of C=C and C=C–H, respectively. These cis-alkene characteristic peaks are due to the unsaturated part of the oil which plays a crucial role in inverse vulcanization reaction [20–23,30,31,41]. Nevertheless, these cis-alkene characteristic peaks disappear in spectra of all obtained copolymers, coupled with the appearance of a new peak representing C–H rocking in the vicinity of a C–S bond, thus confirming the successful reaction of the unsaturated fatty acids with thiyl radicals of the S to produce stable copolymers. Similar results have been found in other edible oil-based copolymers [20,22,23,32,42].

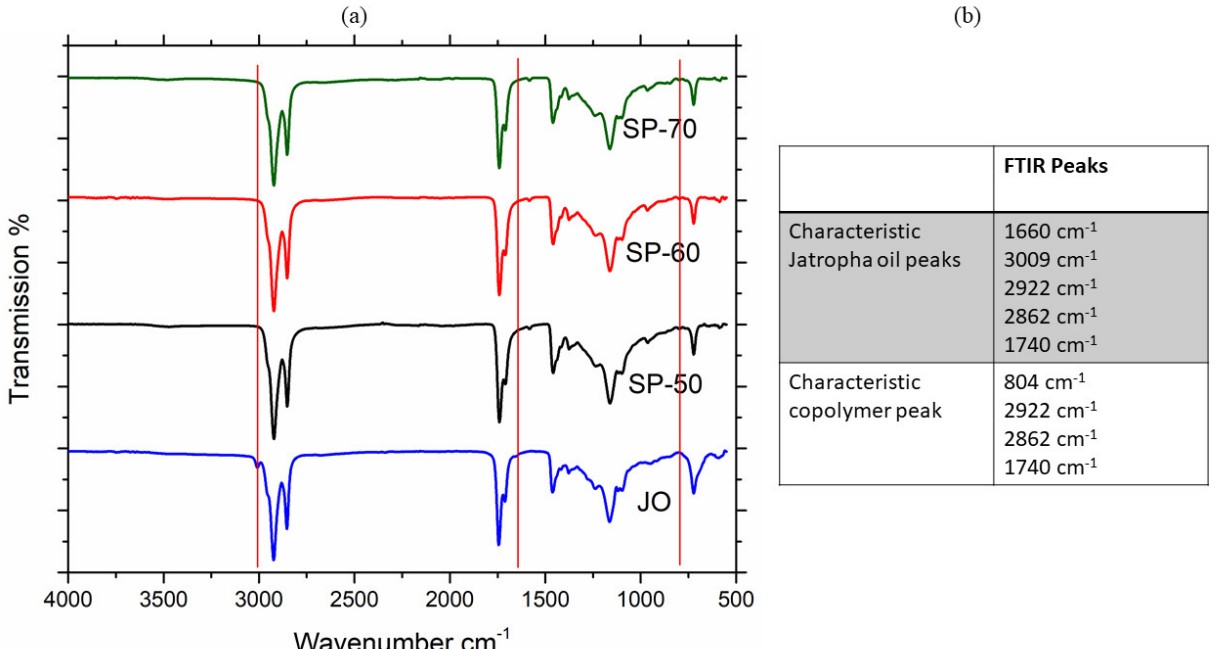

**Figure 1.** (**a**) FTIR-ATR spectra of S polymers with different S loading and Jatropha oil and (**b**) shows the characteristic peaks.

A thermogram of all obtained copolymers and jatropha oil is shown in Figure 2. As can be seen from the figure, the decomposition of oil happens in three steps: in the first step, which start at 289 °C, polysaturated acids start to decompose, followed by monosaturated acids degradation along with remaining polysaturated acids. Oil completely decomposes at around 600 °C [43,44]. Thermograms of copolymers reveal that the obtained copolymers were thermally stable up to 230 °C, which is the highest temperature compared to all reported inverse vulcanized copolymer produced from oil. The degradation happens in a two-step manner: the first step is mainly due to the degradation of the loosely bonded S, followed by the degradation of the oil part of the copolymer and the strongly bonded C–S [20,21,45,46].

Diffractograms of all copolymers along with the elemental S (ES) are shown in Figure 3. The diffractogram of the ES possesses a signal representing the crystalline nature of the $\alpha$-S$_8$ at $2\theta$ of 23, 26, 27.9, and 37.4° [20,30,31]. However, the copolymers were found to be amorphous, as no peak representing the crystallinity of copolymers appeared, except for some less-intense peaks resembling $\alpha$-S$_8$ peaks from 20 to 30 $2\theta$; these peaks are due to the presence of the unreacted S in the copolymer. This was confirmed by investigating the

morphology of the copolymers using the SEM. The SEM images showed the smooth surface of the copolymer along with some isolated S particles, as shown in Figure 4. The amount of these isolated particles increased with increase of the initial S content, thus confirming the presence of the unreacted S in the copolymer and revealing the composite morphology of the polymer.

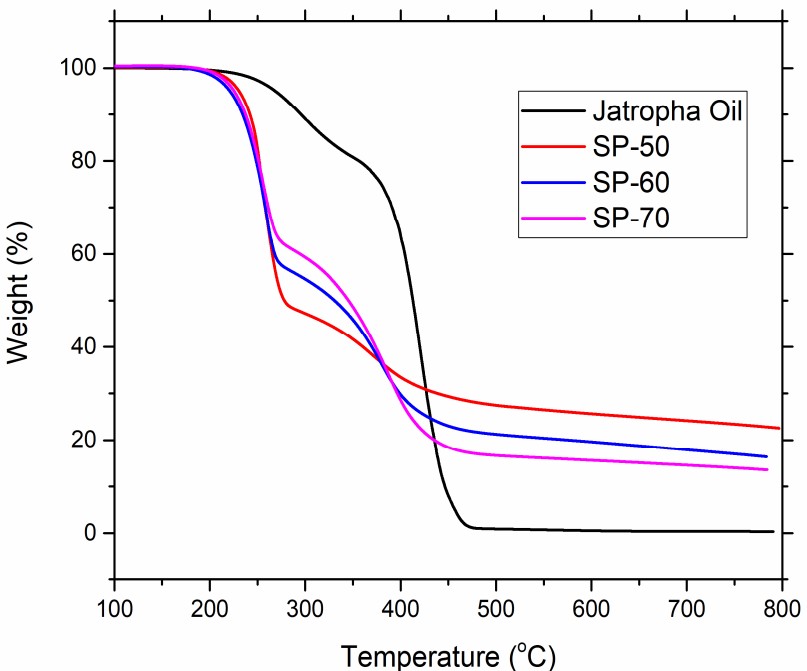

**Figure 2.** TGA thermograms of copolymers and oil.

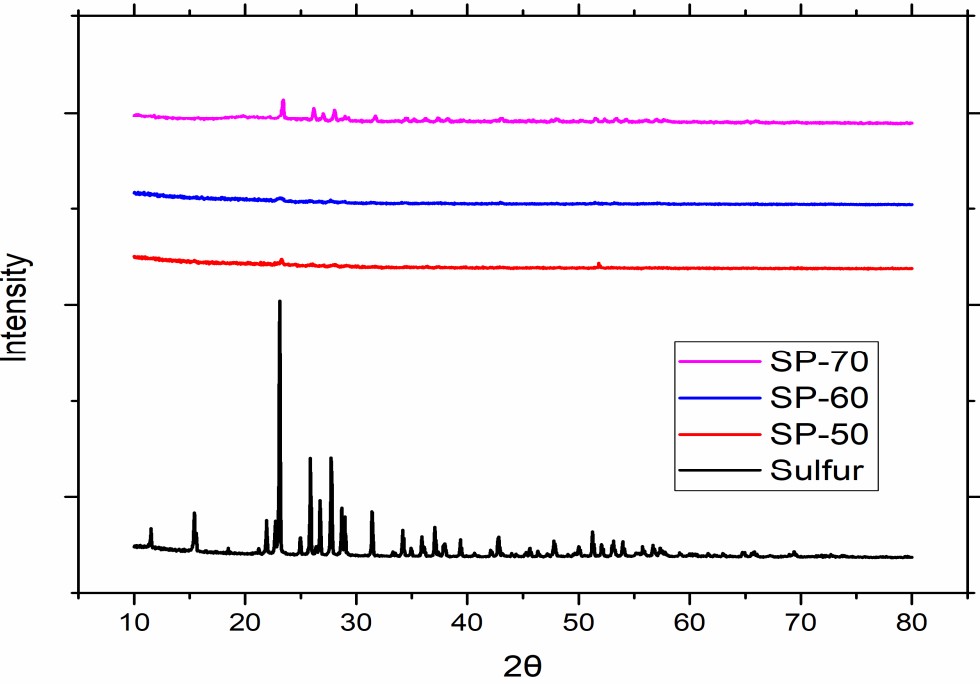

**Figure 3.** p-XRD diffractogram of all copolymer and elemental sulfur.

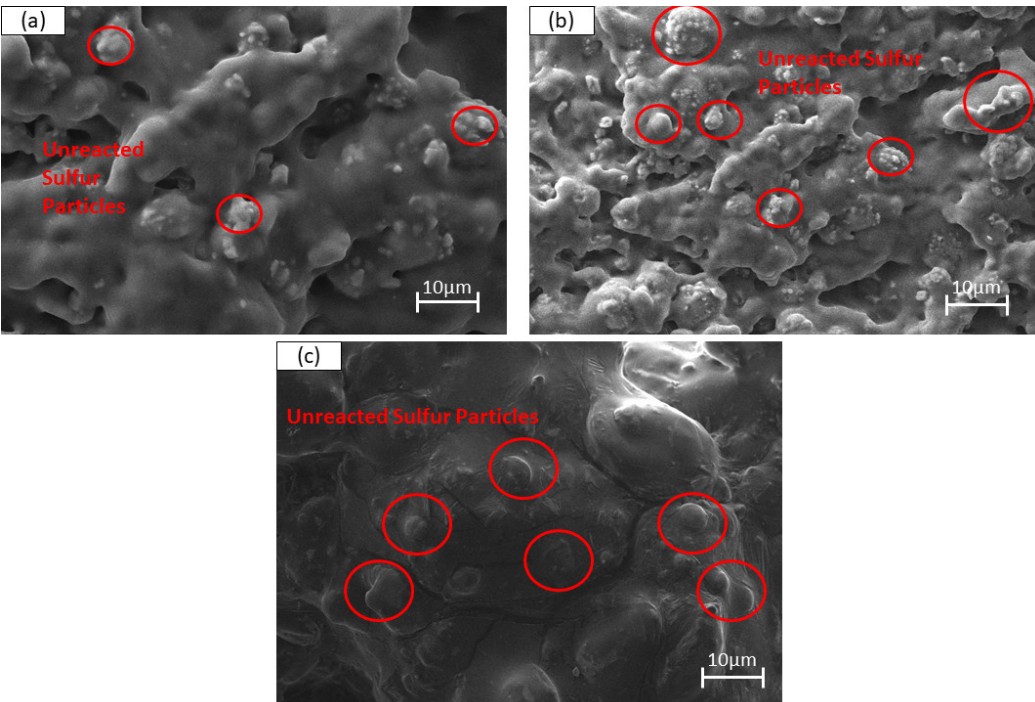

**Figure 4.** SEM images of copolymers with (**a**) 50, (**b**) 60, and (**c**) 70 wt% of S.

FTIR spectra of all SUCFs are illustrated in Figure 5. By comparing the spectra of SUCFs with the copolymer, it was found that spectra of SUCFs possess peaks related to NH symmetrical stretching at 3430–3336 cm$^{-1}$, C=O stretching at 1675 cm$^{-1}$, stretching in NH or NH$_2$ at 1590 cm$^{-1}$, shortening of CN bond at 1460 and 1003 cm$^{-1}$, symmetrical stretching of NH at 1149 cm$^{-1}$, out-phase bending of OCNN at 787 cm$^{-1}$, and bending of NH at 714 cm$^{-1}$ [47]. These peaks are considered characteristics of urea peaks. Appearance of these peaks in spectra of SUCFs and undisturbed peaks of the copolymer thus confirms the successful formation of fertilizer composite.

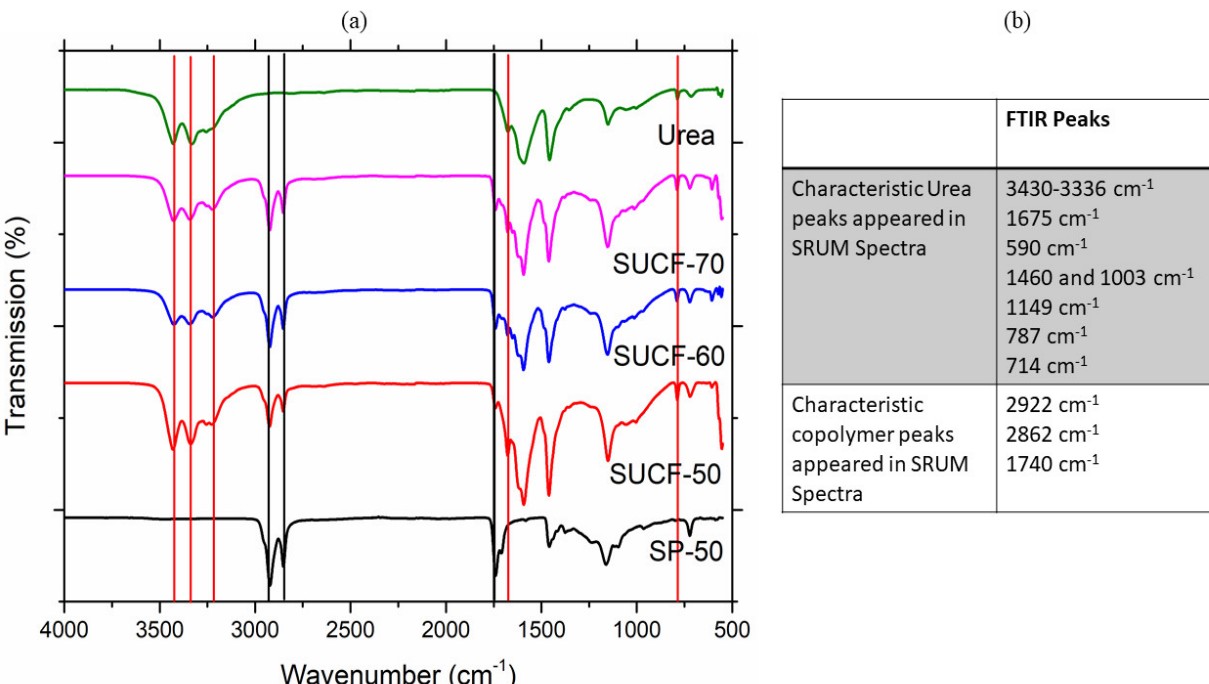

**Figure 5.** (**a**) FTIR-ATR of SUCFS and copolymer and (**b**) characteristic peaks of SUCFS.

To investigate the morphological properties of the SUCFs, SEM images were taken and are depicted in Figure 6. The SEM images of the copolymers obtained at similar conditions depicted in Figure 4 show the smooth surface of the copolymer and isolated unreacted S particles only, but the SEM micrographs of SUCFs shows smooth surface, isolated S particle, and appearance of new isolated particles with different morphology. By conducting the EDX mapping, we found that these new particles represent the urea added to the copolymer, thus confirming the addition of the urea to copolymer structure. The distribution of the urea particle on the surface of the copolymer was found to be nonuniform; this can lead to the sudden release of the nutrient.

The release profile of all SUCFs along with the pristine urea is shown in Figure 7. As can be witnessed from the figure, the N release longevity is highly dependent on the initial S loading of SUCFs as the S loading increases, which results in the increase of the unreacted S leading to micropores generation on the surface of the SUCFs, resulting in quick release of the nutrients. The pristine urea released almost 99.9% of its total N within 24 h, whereas only 13.7%, 25.0%, and 40.0% of total N of SUCF-50, SUCF-60, and SUCF-70 was released. Initial release rate of N from the SUCFs increases with the increase of S loading, which demonstrates the strength of the composites to delay the N release; the lesser the S content, the stronger it will be to prevent the N loss. SUCF-50 outperformed all SUCFs in term of N release longevity, as it released only 70% of its total N after 16 days of incubation which is far better than the other biopolymer-based SRFs, which released 100% of their N in less than 100 h. At some points, SUCFs showed burst release of N; for example, on the 5th day of the incubation, it suddenly released 28% of the N: this happened because of the nonuniform distribution of urea on the surface of the copolymer, as shown in SEM images. This can be controlled by solvent-assisted mixing of the copolymer with urea. SUCF-50 perfectly follows the European Standard (EN 13266, 2001) which states that slow-release fertilizer should not release more than 15% of its nutrients in 24 h.

Release of the N from the best-performing SUCF was also tested in soil, and the obtained results are presented in Figure 8. The release of the N in soil was observed to be even slower than the release in distilled water. SUCF-50 released only 35% of its total N in 20 days. The release of the N in its first 20 days of incubation was slow, and it can be regarded as a lag period. The soil contained *Aspergillus niger,* which oxides the S backbone of the copolymer to sulfate because of the microorganism process. Thus, this fertilizer not only delayed the release of the N, but could also produce sulfate, which is considered as a secondary, but indispensable, nutrient required for the plant's growth [32,48].

The results of copolymers biodegradability in soil are depicted in Figure 9. The weight loss of all copolymers increased with the increase of soil incubation period. The SP-50 copolymer showed the lowest weight loss on the 4th day of the incubation. The weight loss reached 26% on the 48th, 28th, and 20th day of incubation for SP-50, SP-60, and SP-70, respectively, demonstrating that the copolymer was degraded in soil, but it will take longer to fully decompose. SP-60 and SP-70 samples showed high error percentages which might be due to the presence of a high amount of unreacted elemental S in the copolymer structure causing a probable loss in the soil or during washing. Microorganism attack on the surface of the copolymer which oxidizes the S chain to sulfate results in weight loss, making this copolymer biodegradable in nature, which is an additional benefit of using this material as coating material for urea, as this mitigates the problem of pollution caused by the coating shell left in the soil after release of nutrients.

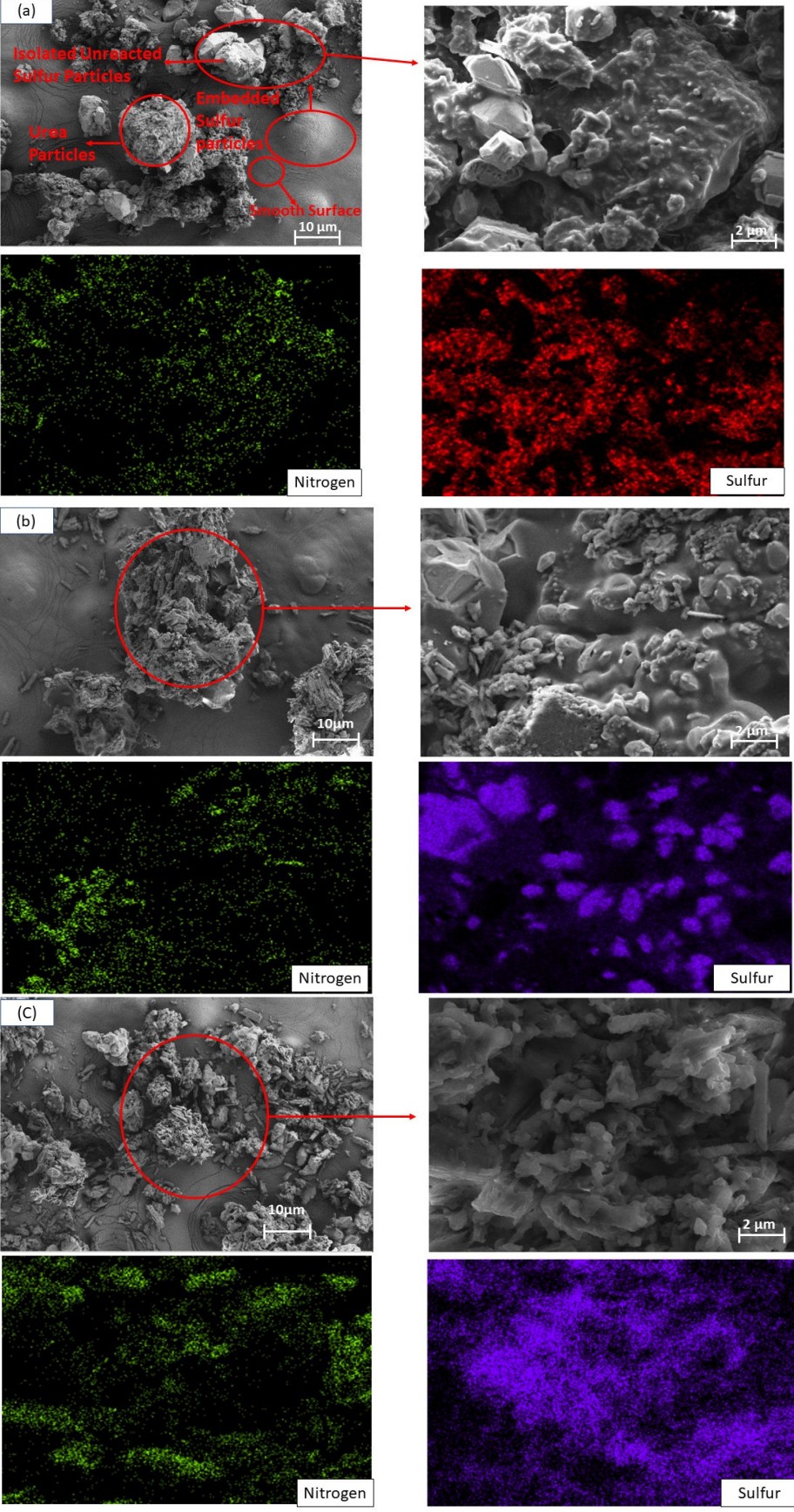

**Figure 6.** SEM micrographs of SUCFs along with EDX mapping with (**a**) 50, (**b**) 60, and (**c**) 70 wt% S.

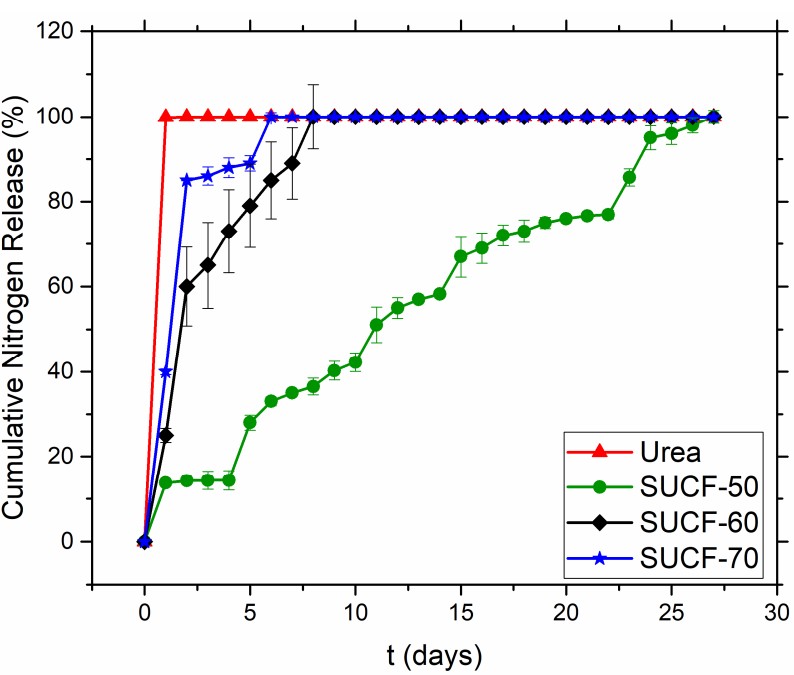

**Figure 7.** Cumulative nitrogen release vs. time (distilled water).

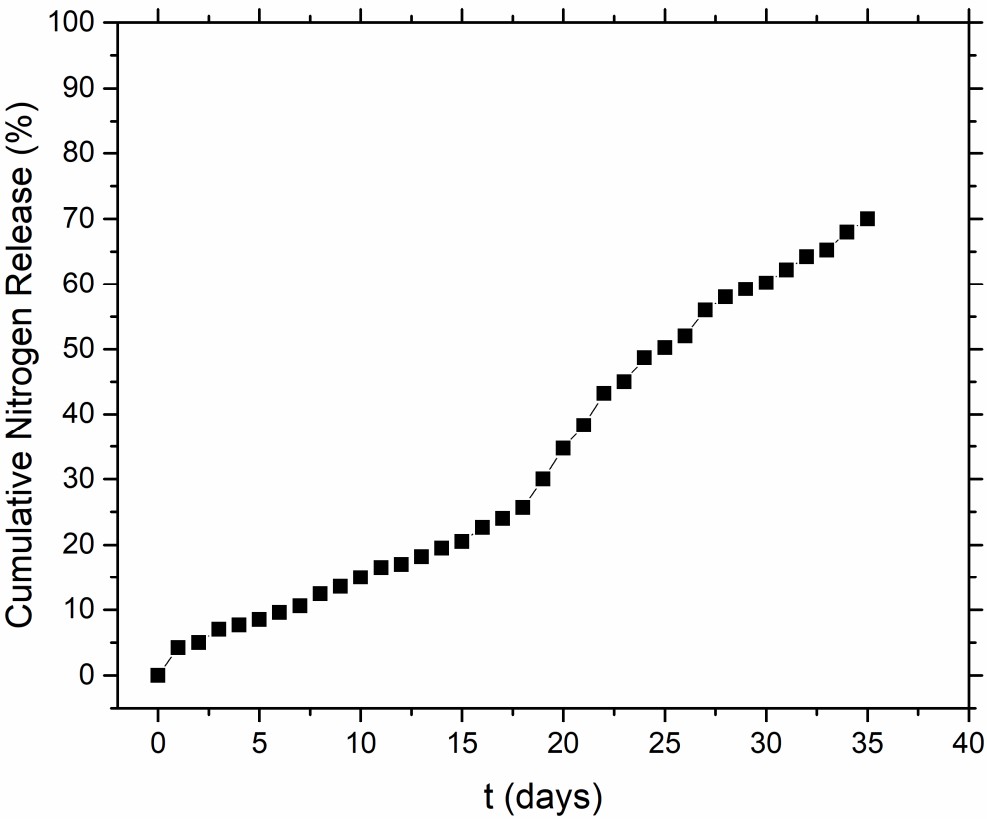

**Figure 8.** Cumulative nitrogen release in soil over time by the SUCF-50 fertilizer.

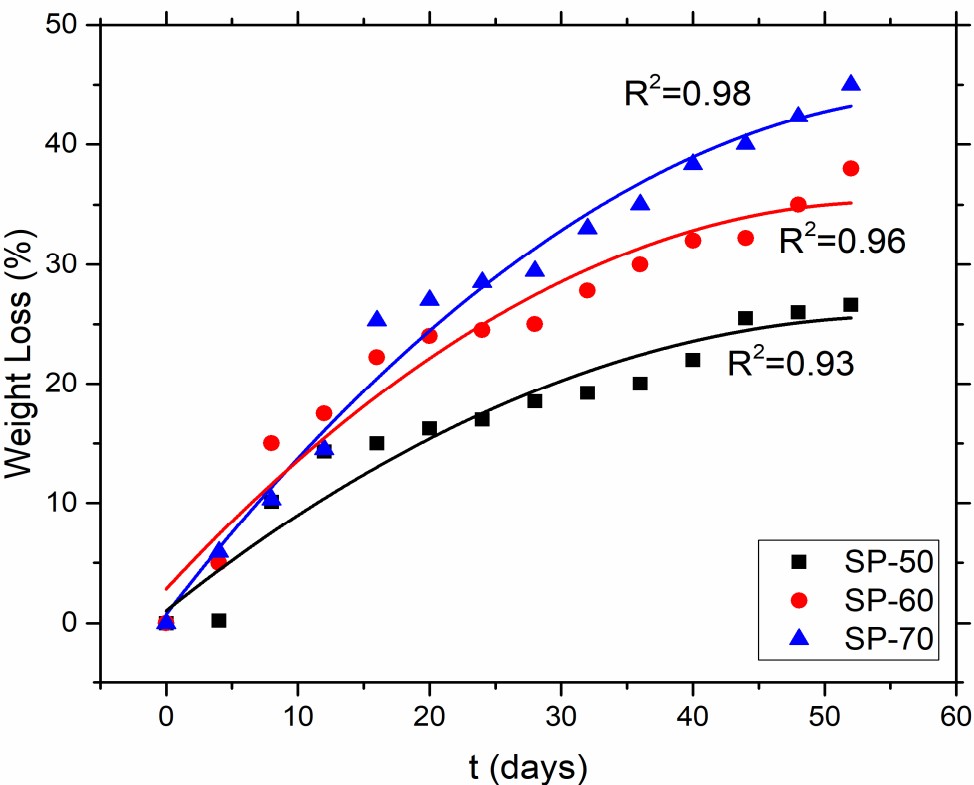

**Figure 9.** Weight loss of copolymers vs. soil burial time (days).

## 4. Conclusions

A novel, facile, solvent-free methodology was adopted to synthesize SRF using inverse copolymers. These copolymers were prepared by reacting jatropha oil and S at 170 °C. Slow-release urea fertilizers were instead produced by mechanically mixing copolymers with urea. The best performance was observed with the fertilizer developed using copolymer with 50 wt% S, as it releases only 70% of total N in distilled water after 16 days of incubation and 35% in soil after 20 days of incubation. All copolymers were biodegradable in the soil. Although obtained results need to be confirmed in real open-field conditions, they clearly suggest that inverse copolymers can be used to better synchronize nutrients release and plants uptake.

**Author Contributions:** Conceptualization, A.S.M.G. and M.M.N.; methodology, A.S.M.G., R.S. and A.A.; software, A.S.M.G. and O.U.R.; validation, A.S.M.G. and C.M.; formal analysis, A.S.M.G., R.S., A.A.R. and M.I.H.; investigation, A.S.M.G., O.U.R. and A.A.R., resources, R.S. and M.M.N., data curation, M.M.N., C.M. and M.I.H., writing—original draft preparation, A.S.M.G. and C.M., writing—review and editing, A.S.M.G., R.S., C.M. and O.U.R.; visualization, A.S.M.G. and O.U.R., supervision, R.S. and M.M.N., project administration, R.S.; funding acquisition, R.S., M.M.N. and C.M. All authors have read and agreed to the published version of the manuscript.

**Funding:** This research was funded by [Murata Science Foundation] grant number [015ME0-2369].

**Data Availability Statement:** The data is no available as this data will be used for future study.

**Acknowledgments:** This research was carried out under a research grant (015ME0-239) from Murata Science Foundation.

**Conflicts of Interest:** The authors declare no conflict of interest.

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
