# Peer review of "Degradable Slow-Release Fertilizer Composite Prepared by Ex Situ Mixing of Inverse Vulcanized Copolymer with Urea"

_agronomy, doi:10.3390/agronomy12010065_

Round 1

Reviewer 1 Report

In this work, Slow-release Urea fertilizer was developed using Copolymer synthesized via inverse vulcanization of jatropha oil. Characterization of obtained copolymer was analyzed in detail using FTIR, TGA, XRD and SEM. Whereas, characterization of obtained SUCF was conducted by FTIR and SEM. Nitrogen release test was done by immersing SUCF in distilled water and burying in soil. However, the present manuscript is severely plagued by problems in the presentation, many of the presented trends are exposed with quite lack interpretation of data. The manuscript requires major revisions before being considered for publication in agronomy or any journal in this field.
1.    The author conducted a TGA test on the obtained polymer composites, is there a correlation between the characterization obtained from the TGA test and its use as a matrix on SUCF?
2.    As shown in Figure 2 (TGA), It may predict that the residual substance is sulfur (see at temperature around 800oC). However, SP60 has higher S residue that both SP50 and SP70. Please add the explanation of the obtained data.
3.    Does the presence of sulfur affect the acidity (pH) both soil and water during the Nitrogen release? It is required the pH measurement to ensure that S in matrix is not detaining the plant growth.
4.    Author stated that SUCF with inverse vulcanized copolymer as matrix has better biodegradation and nutrient release longevity. However, this manuscript is not confirming the degradability and its release longevity. 
-. It is better added the comparison of SUCF morphology (SEM) before and after N release test. From this comparison, we can analyze  does matrix degradation/decomposition accompany the nitrogen release? 
-. Add a table tabulated the nitrogen release duration from other reports, thus we get depiction of the obtained SUCF.
4. Interpretation of figure 3 (XRD) (line 202, page 6) should present the quantitative crystallinity of copolymer. Thus, we can analyze the effect of S in composite. Generally, sulfur acts as a crosslinker in polymer. From this, it can be concluded the relation between crystallinity of composite and N release duration
5. The presence of Sulfur ( 50,60 and 70%) in composite shows the unreacted sulfur. It may indicate the S excess. It requires additional data to confirm the maximum S addition so that the composite does not have unreacted Sulfur.

Reviewer 2 Report

Studies on slow-release urea fertilizers prepared with free solvent method have been a significant research subject. This work was based on Inverse Vulcanized Copolymer as the raw material to achieve the purpose of the subject. The article introduced the preparation of Inverse Vulcanized Copolymer and SUCF slow-release fertilizers. And, structural characterizations including SEM, TG, FTIR for the prepared SUCF slow-release fertilizer were carried out. The slow-release behavior in water and soil and the biodegradability of slow-release fertilizers were investigated and discussed. The research was interesting. But this work currently needs further improvement in writing before it can be considered.

  1. line 69, The research background of Inverse Vulcanized Copolymer is very important for this work, but in the introduction part, its introduction was too simple, even less than that of SRF hydrogels.

  1. The methods part should be rewritten.For example, line120, (1) What does ex-situ mixing mean? (2) "SRF composited were synthesized by ex-situ mixing of produced copolymers with powdered urea using mechanical mixing at a speed of 150 rpm to get matrix with uniform distribution of the N/urea. The mixing was done immediately after removing the glass Vial from heating after 1 hour for 10 min." What is the meaning of the sentence, it is not clear. What is the heating temperature? How to control the temperature?

  1. Many grammar and writing errors! For example,line201-204, "However, the copolymers were foundto be amorphous as no peak representing the crystallinity of copolymers appeared expected some less intense peaks resembling with α-S8 peaks from 20 to 30 2θ these peaks are due to the presence of the unreacted S in the copolymer. Which confirmed by investi-gating the morphology of the copolymers using SEM. ”

  1. FTIR discussions about copolymers(Line 171-184)and  SUCFs (line214-220)should be written together .
  2. Fig6, the specific data of EDX should be providedfor knowingwhich particles was S particles or urea particles.

6.In Fig7, SUCF-50 has the best release behavior in  SUCF-50,sucf-60 and sucf-70. But only 50% S is incomplete. why not continue to reduce the content of S in SUCFs? That is, it was necessary to study release behavior SUCF-40, SUCF-30, SUCF-20... until the inflection point appear?

  1. Line 259-261, in figure 8, the 35thday release behavior does not seem to have reached a constant release.

Reviewer 3 Report

The authors have reported the improvement the crop yield and nutrient uptake efficacy, a novel
slow-release urea composite fertilizer (SUCF) was developed using inverse vulcanized copolymer
with better biodegradation and nutrient release longevity. Copolymers were synthesized via inverse
vulcanization of jatropha oil; and their properties were evaluated using thermogravimetric analysis
(TGA), Fourier Transform Infrared Spectroscopy (FTIR), powdered-X-Ray diffractometer (p-XRD)
and scanning electron microscopy (SEM). SUCFs were developed by ex-situ mixing of inverse
vulcanized copolymer with urea powder using mechanical mixer and their properties were
evaluated using FTIR and SEM. Nutrient release longevity of developed fertilizers was evaluated
in both soil and distilled water. The leaching test revealed that only 70 % of the total nutrients of
SUCF prepared from 50 wt% sulfur copolymer after 16 days of incubation in distilled water whereas
it releases only 30 35 % nutrients after 20 days in soil. The biodegradability of all copolymers
developed was investigated by burying in soil; and it revealed their biodegradable nature as weight
loss was observed; which increased with the increase of incubation period.

The composition of the manuscript is logically adequate. In principle, abstract reflects the general
items and the intentions of the authors. The introductory section provides a relatively
comprehensive background of the topics that immediately allows the reader to acquaints with the
principal trends of the preparation of the slow-release urea composite fertilizer (SUCF). All figures
and tables are quite acceptable. The text and terminology are transparent and understandable for
the experts in the slow release urea fertilizers area. Below some minor questions:
1-What is the composition of jatropha oil?
2-Why the % of S used to prepare the copolymer is higher (50 to 70 %)?
3-What corresponding red-color solution obtained during the Nitrogen release test in distilled
water?
4- In the TGA thermograms, the name of Y-axis is Weight % and not Weight loss % as 100 °C
the weight loss is zero.
5- Which kind of interaction between copolymer and urea during the fabrication of fertilizers?

Reviewer 4 Report

  1. Please correct literature citation and references as required in the journal.
  2. Please correct language and editing errors, g. fertilizers (L. 49), fertilisers (L. 51), 2.2.2.4Scanning (L. 115), 2g (L. 163), 167 oC (L. 167), the font size in figures, etc.
  3. Please add references to the all methods and calculations used.
  4. Parameters should be subjected to statistical analysis to determine their statistical validity.

Round 2

Reviewer 1 Report

The authors have revised and added interpretation of the data as suggested previously 

Reviewer 2 Report

I think the author has made modifications according to my requirements, and this revised manuscript is suitable for publication.